# Neural Signatures of Error Processing in Depressed Adolescents with Comorbid Non-Suicidal Self-Injury (NSSI)

**DOI:** 10.3390/biomedicines10123188

**Published:** 2022-12-08

**Authors:** Kathrin Malejko, Stefan Hafner, Rebecca C. Brown, Paul L. Plener, Georg Grön, Heiko Graf, Birgit Abler

**Affiliations:** 1Department of Psychiatry and Psychotherapy III, University Hospital Ulm, 89075 Ulm, Germany; 2Department of Child and Adolescent Psychiatry and Psychotherapy, University Hospital Ulm, 89075 Ulm, Germany; 3Department of Child and Adolescent Psychiatry, Medical University of Vienna, 1090 Vienna, Austria

**Keywords:** response inhibition, commission errors, error processing, major depression, non-suicidal self-injury (NSSI), adolescence, Go/No Go

## Abstract

Non-suicidal self-injury (NSSI), as a highly prevalent psychiatric symptom in adolescents and young adults, is defined as the deliberate destruction of body tissue without suicidal intent. Impulsivity and dysfunctional response inhibition have been suggested to play a central role in adolescents’ vulnerability to self-harm. To investigate the potentially distinct neurobiology of NSSI, we used a well-established Go/No Go task in which activation of the inferior frontal gyrus (IFG) and dorsal anterior cingulate cortex (dACC) is interpreted as a neural correlate of processing failed response inhibition. Task-based functional magnetic resonance imaging data were obtained from 14 adolescents with a diagnosis of major depression and a history of NSSI (MD-NSSI), 13 depressed adolescents without NSSI (MD-only), and 14 healthy controls (HC). In line with hypotheses of dysfunctional response inhibition, we observed increased rates of commission errors in MD-NSSI along with significantly reduced error-related activations of the dACC and IFG. Intact response inhibition, as reflected by low commission error rates not different from HC, was observed in MD-only, along with increased activation of the error-processing network. Our findings support the hypothesis of a distinct neurobiological signature of NSSI. Further research on biomarkers of NSSI could focus on behavioral and neural correlates of failed response inhibition.

## 1. Introduction

Non-suicidal self-injury (NSSI) is defined as the deliberate destruction of one’s own body tissue without suicidal intent [1]. The most common methods are cutting, scratching, hitting or banging, carving, burning and scraping [2]. NSSI is particularly prevalent in mid-adolescence [3], with an estimated 23% lifetime prevalence among adolescents in community samples [4]. In adolescent psychiatric samples, prevalence rates of NSSI have found to be 50–60% for one incident or repetitive NSSI [5]. NSSI is observed by itself, but is seen frequently comorbid to further psychopathology and in the broad context of psychiatric disorders, including mood disorders [6,7,8]. NSSI has been categorized as impulsive behavior. Impulsivity and self-harm are also core diagnostic criterions of borderline personality disorder. However, in earlier adolescence, it has been shown that, in particular, mood disorders and NSSI often co-exist [9,10]. While NSSI is seen as an important predictor for the development of borderline personality disorder in early adulthood, only a minority of adolescent patients with self-injurious behavior fulfil the criteria for the diagnosis around the onset of NSSI behavior [11].

NSSI serves the primary function of diminishing negative feelings or thoughts in the majority of cases [12]. After engaging in NSSI, relief from these, and even pleasant feelings or thoughts, has been reported [8]. Low planning ability and mood-based impulsivity-related traits have been consistently associated with an increased risk of developing NSSI [13]. Thus, impulsivity might play a central role in adolescents’ vulnerability to self-harm [14] and has been identified as a risk factor [15,16]. Using several assessment methods, Fikke et al. [17] were able to show impaired inhibitory control in a subgroup of adolescents with NSSI of low frequency. Individual differences in impulsivity have been suggested to also relate to individual differences in cognitive control [18], which in turn has been linked to inhibition as one of its essential components [19]. Response inhibition is the cognitive process required to cancel an intended action and is a widely studied operationalization of inhibition [20].

Response inhibition and its neurobiology can be tested using the well-established combined Go/No Go Eriksen flanker paradigm [21]. Here, subjects are instructed to perform speeded responses on Go trials (e.g., pressing a button in response to the letters P, V), while not responding on No Go trials (e.g., to the letter R, U). The established index of inhibitory control measured hereby is the number of errors that a subject commits on No Go trials. A higher number of commission errors can be interpreted to relate to lower inhibitory control and is suggestive of greater levels of impulsivity. Besides the behavioral findings, the task has been successfully implemented in functional magnetic resonance imaging (fMRI) settings and thus permits investigation of the underlying neurobiology [22,23,24,25,26]. 

To date, several neuroimaging studies suggest that unsuccessful response inhibition indexed by commission errors in No Go trials activates prefrontal cortex regions, the supplementary motor area (SMA, e.g., [24,25]), and especially and consistently the right inferior frontal gyrus [19,27,28,29]. Functional and structural MRI [30,31,32,33] and EEG data [34] brought evidence that damage in this region leads to permanent impairments in subjects’ response inhibition. 

Intending to shed light on the neurobiology of NSSI, we hypothesized that adolescents with NSSI would show altered neural reactivity of prefrontal cortex regions during a task examining response inhibition. In particular, the inferior frontal gyrus and anterior cingulate cortex should play a significant role. To control for the effects of comorbid psychopathology, adolescents with at least five episodes of NSSI during the past year and a diagnosis of major depression were compared to depressed adolescents without a recent history of NSSI. We used the modified Erikson flanker Go/No Go paradigm without incentive manipulations, which has been shown to reliably elicit neural responses in brain areas processing failed response inhibition [22,23,24,25,26]. Based on previous electrophysiological and neuroimaging studies [24,35,36,37,38], we expected a blunted neural response in brain regions previously related to response inhibition and error processing in patients with comorbid NSSI. By contrast, depressed adolescents without NSSI were expected to show the previously reported [24] unimpaired activation of the error-processing network. 

## 2. Materials and Methods

### 2.1. Participants

Twenty-eight adolescents fulfilling the diagnostic criteria for major depression (MD) were recruited for this study. Of these, 14 met the criteria for comorbid non-suicidal self-injury (‘MD-NSSI’ group; 11 female, 3 male). According to Section 3 of the DSM-5 [1], NSSI in this group was reported at least five times within the past year (mean number of NSSI events within the past year: 93.3, SD = 225.9). By contrast, the other 14 adolescents showed no NSSI events within the last 12 months (‘MD-only’ group; 11 female, 3 male) great enough to fulfil diagnostic criteria. Five participants of the MD-only group reported minor lifetime NSSI (two participants once, one participant four times, and two participants seven times, all events at least one year before the onset of the study). Additional symptom-based diagnoses according to DSM-IV at the time of the study were assessed using the German version of the semi-structured interview Kiddie-SADS-PL (Schedule for Affective Disorders and Schizophrenia for School-Age Children Present and Lifetime [39]), and revealed in both groups (MD-only/MD-NSSI) anxiety disorder (n = 2/n = 1), posttraumatic stress disorder (n = 4/n = 0), eating disorders (n = 2/n = 2), borderline personality disorder (n = 0/n = 1), and ADHD (n = 1/n = 2). Recruitment took place in the inpatient and outpatient units of the University Hospital for Child and Adolescent Psychiatry and Psychotherapy Ulm (Germany) and in a private medical practice for child and adolescent psychiatry in Ulm. Fifteen healthy adolescent controls (‘HC’ group; 12 female, 3 male) with no clinical diagnoses and no current or past psychiatric disorders were recruited as a control group. Dropouts included one female participant in the MD-only group and one female participant in the HC group, who both interrupted the fMRI procedure. Of all 41 participants, two were left-handed (HC: n = 0; MD-only: n = 1; MD-NSSI: n = 1). Two participants (MD-only: n = 1; HC: n = 1) reported regular cigarette smoking, which was forbidden at least two hours before fMRI scanning. None of the participants reported excessive use of alcohol, as assessed by the corresponding section of the Kiddie-SADS-PL. Other substance use disorders, bipolar disorder, schizophrenia, current medical illnesses, or a history of epilepsy were exclusion criteria. All participants had reached puberty. All female participants were scanned between day one and ten of their menstrual cycles or after at least 14 days of continuous intake of oral contraception.

Both patient groups (MD-only and MD-NSSI) were matched for age, gender, IQ, and depression scores (see Table 1). Psychotropic medication was kept stable within patient groups for at least two weeks before fMRI scanning to ensure steady-state conditions and included SSRIs (n = 7), mirtazapine (n = 2), and methylphenidate (n = 1). Healthy controls were average-matched to patient groups regarding age, gender, and IQ. Nevertheless, the mean age of the HC group was one year less than that of the patients’ average. The study was approved by the Institutional Review Board of Ulm University, Germany (EA 58/2012. Written informed consent was obtained from legal guardians and participants. All procedures were performed according to the Declaration of Helsinki.

### 2.2. Psychometric Measurements

Past and current non-suicidal self-injurious behavior was assessed by the validated [40] semi-structured Self-Injurious Thoughts and Behaviors Interview (SITBI [41], German version by [40]). Present depressive symptoms were measured with the German version of the Beck Depression Inventory, second edition ([42]; German version by [43]), and the semi-structured interview Children’s Depression Rating Scale—Revised ([44]; German version by [45]). Intelligence quotients (IQs) were either estimated by the Prüfsystem für Schul und Bildungsberatung für 6.-13. Klassen (PSB) [46] or by using the Wechsler Intelligence Scale for Children (WISC-IV; Wechsler [47]). Parasuicidal behaviors were assessed with the semi-structured Self-Injurious Thoughts and Behaviors Interview—German (SITBI-G; [40]). 

### 2.3. fMRI Procedure

The data presented here were collected as part of a larger experimental setup where participants first underwent a Cyberball task [48] and unpleasant electrical stimulation. Findings have already been reported [49]. 

To examine error processing during fMRI, a combined Go/No Go Eriksen flanker paradigm [21] was performed. The task has been demonstrated to be suitable to examine error monitoring and response inhibition in various electrophysiological and functional MRI studies [22,23,24,25,26]. As reported previously [23,24,25], five-letter strings with letters R, U, P, and V were shown. The action-relevant target was always mid-standing. During Go trials, subjects had to respond to the target letter R with their right index finger on a two-button box and to the target letter U with their right middle finger. In No Go trials, subjects were instructed to withhold their response whenever the letters P or V appeared. Combinations of target and flanker stimuli were either congruent (all five letters the same) or incongruent. In incongruent Go trials, targets flanked by visually similar No Go targets were shown (e.g., VVUVV). In incongruent No Go trials, visually similar Go targets flanked the central No Go target (e.g., UUVUU) (see Figure 1). After stimulus presentation, a response time window depending on individual reaction times was established. Feedback (German expressions for either correct of wrong) followed the response. Upon delayed responses, the feedback “faster” was displayed. To secure enough incongruent No Go errors, participants were instructed to emphasize speed over accuracy in Go trials. Incorrect button pushes and overly slow responses were recorded as separate types of errors. Overly slow responses were not further evaluated. Individual reaction time windows were determined during a training session before scanning. For each combination of the factors condition (Go, No Go) and type (congruent, incongruent), 66 trials were prepared (264 trials in total). The duration of one trial was 1.9 s, the mean inter-trial interval 3.01 s. The average stimulus-onset asynchrony (SOA) was 19.5 s for events of the same combination of condition by type (e.g., incongruent No Go). The entire fMRI task lasted around 22 min. A standard personal computer was used to register reaction times and the correctness of the subjects’ responses per trial.

### 2.4. Functional Data Acquisition

fMRI data were acquired using a 3.0 T MAGNETOM Allegra Scanner (Siemens, Erlangen, Germany). A T2*-sensitive gradient echo sequence was used for functional imaging, with an echo time (TE) of 33 ms, a flip angle of 90°, a field of view (FOV) of 230 mm, and a slice thickness of 2.5 mm, with an interslice gap of 0.5 mm. At a repetition time (TR) of 2000 ms, 35 transversal slices were recorded with an image size of 64 × 64 pixels during the Go/No Go task. Anatomical high-resolution T1-weighted images (1 × 1 × 1 mm voxels) were obtained (bandwidth (BW) = 130 Hz/Pixel, TR = 2500 ms, TI = 1.1 s, echo time (TE) = 4.57 ms, flip angle 12°) to co-register and normalize data into a standardized stereotactic space.

### 2.5. Data Analysis

#### 2.5.1. Behavioral Data and Psychometric Measurements

Statistical analyses were calculated with JASP [50]. Excel (Microsoft Corporation (2022)) was used to create the figures presented. Between-group differences in non-fMRI data were assessed with one-factorial ANOVAs. Post-hoc analyses were Bonferroni-corrected. The significance level was set to *p* ≤ 0.05. One participant in the MD-only group did not commit any errors of commission and could therefore not be included in the fMRI analyses on the contrast ‘incorrect minus correct incongruent No Go trials’. 

#### 2.5.2. fMRI-Data

We used Statistical Parametric Mapping (SPM12, Wellcome Department, London, UK) under MATLAB R2020a (Math-Works, Version 9.8.0.1323502, Natrick, MA [Computer Software]) to pre-process imaging data from each session, including realignment, slice timing, and normalization into the standard MNI (Montreal Neurological Institute) template, with a spatial resolution of 2 × 2 × 2 mm^3^. Spatial smoothing was performed using an 8 mm full width at half maximum (FWHM) Gaussian kernel. Low-frequency drifts were removed via high-pass filtering. Intrinsic autocorrelations were accounted for by an AR (1) model. On the first level, we modeled individual event types as trains of delta functions at stimulus onset and convolved them with the canonical hemodynamic response function. Regressors for individual fixed-effects analyses were obtained by modeling individual events for the combinations of factors condition (Go, No Go), type (congruent, incongruent), and within-time responses (correct, incorrect). This resulted in eight conditions of interest. Out-of-time responses and missed Go events were also modeled, but as conditions of no interest. The six realignment parameters were added to the design matrix. Of note, congruent Go and No Go trials did not yield errors consistently across subjects. Thus, the ‘congruent’ conditions, though modeled, were not included in second-level analyses.

For second-level group analyses, we computed a 3 × 4 ANOVA model with factors ‘group’ (HC, MD-only, MD-NSSI) and ‘condition’ (Go/incongruent/correct, Go/incongruent/incorrect; No Go/incongruent/correct, No Go/incongruent/incorrect). In line with previous analyses [23,24,25], hypothesis-driven group differences were calculated for the contrast of incorrect minus correct incongruent No Go trials. To exclude the effects of psychostimulant MPH medication, we furthermore recalculated the group comparisons after excluding the respective participant (n = 1). Results were considered significant at a threshold of *p* ≤ 0.005 at the voxel level, and *p* ≤ 0.05 at the cluster level, with a minimum of 183 contiguously significant voxels.

## 3. Results

### 3.1. Demographic Data and Psychometric Measurements

Depression scores in both patient groups were significantly higher relative to HC. An ANOVA for age revealed that HC were, on average, one year of life younger than the patient groups. However, this difference was significant only when HC subjects were compared to the MD-only group. According to the SITBI-G [40], one patient in the MD-only group and four patients in the MD-NSSI group reported suicide attempts, with a total number of one in MD-only and a mean number of 3.3 in MD-NSSI (SD = 3.9), respectively. Suicide plans were reported by the same patient in the MD-only group on one occasion within the past year. In the MD-NSSI group, six patients reported suicide plans at a mean rate of 3.3 (SD = 2.8) episodes within the last year. None of the participants in the HC group reported suicide attempts or plans. Suicidal thoughts were reported by participants from each group (HC: n = 2; MD-only: n = 12; MD-NSSI: n = 13) at mean rates from one episode within the past year (HC group), over 2.8 (SD = 3.0; MD-only), to 13.8 (SD = 31.1; MD-NSSI) episodes in the past year. The number of errors in Go trials did not differ significantly between groups. However, for No Go trials, a significant effect of group was found. Post-hoc tests confirmed a significantly greater number of commission errors in MD-NSSI compared to HC, while, in MD-only, the occurrence of commission errors was not significantly different from that in HC. Furthermore, reaction times showed a significant effect of group for incongruent Go trials. For both correct and incorrect Go trials, MD-only and MD-NSSI showed faster reaction times relative to HC. Table 1 summarizes demographical data and psychometric measurements. Table 2 summarizes task performance data obtained from the fMRI experiment.

### 3.2. fMRI Data

We first analyzed differential neural activations corresponding to errors (incorrect minus correct) in incongruent No Go trials separately for each group (HC, MD-only, MD-NSSI). In line with previous investigations [23,24,25], the following brain regions showed reliable differential neural activations for this contrast in each group: dorsal anterior cingulate cortex (dACC), supplementary motor area (SMA), bilateral anterior insula, and bilateral inferior frontal gyrus (IFG). More details are provided in the Appendix A (see Appendix A).

#### 3.2.1. Incorrect Minus Correct Incongruent No Go Trials

Investigating the effect of NSSI on neural error processing (incorrect minus correct incongruent No Go trials) by comparing the two patient groups revealed lower differential neural activations of dACC, left IFG (pars triangularis and pars opercularis), left supramarginal gyrus, and of the left inferior parietal cortex in MD-NSSI relative to MD-only (see Table 3 and Figure 2A,B). We found no significantly higher differential neural activations in MD-NSSI compared to MD-only. 

Comparing error-related neural activation between each patient group and HC showed relatively increased left-lateralized differential activation in the MD-only group: left anterior insula, left postcentral gyrus, left inferior parietal cortex, left precuneus, and left cerebellum. Inverting the contrast (HC > MD-only group) did not yield any significant results. MD-NSSI patients showed reduced differential activation of the right middle frontal gyrus, right IFG (pars triangularis and pars opercularis), right supramarginal gyrus, and left inferior parietal cortex relative to HC. Significantly higher differential neural activation in MD-NSSI compared to HC was found for the left precuneus (see Table 3). 

To confirm that our findings were not driven by confounding motor activity (button press upon incorrect No Go trial), we calculated the contrast ‘incorrect minus correct incongruent Go trials’ with a motor component present in each condition as a conjunction across all groups (HC, MD-only, MD-NSSI). This statistical map was then used as an inclusive mask thresholded at *p* ≤ 0.05, uncorrected for the re-calculation of the above between-group comparisons for failed response inhibition. All previous clusters could be confirmed at a minimum cluster size of 135 voxels. After excluding the participant receiving psychostimulant medication, all significant clusters from previous group comparisons were confirmed at a minimum cluster size of 95 voxels. Thus, both control calculations yielded similar results, with only smaller cluster sizes compared to the original analyses. 

In summary, we revealed neural activations of brain regions reliably linked to neural error processing. MD-NSSI patients showed reduced neural error processing in both hemispheres relative to MD-only patients and healthy controls. The MD-only group showed enhanced neural error signaling when compared against the group of MD-NSSI and HC.

#### 3.2.2. Incorrect Minus Correct Incongruent Go Trials

We found robust activation in the error-processing network comprising dACC and bilateral IFG in each of the three groups for the contrast ‘incorrect minus correct incongruent Go trials’ (see Appendix A).

Investigating differences between the two patient groups in error-related brain activation for Go trials, we observed significantly lower neural activations of right and left IFG for the contrast ‘incorrect minus correct incongruent Go trials’ for MD-NSSI. This was similar to the effects in No Go trials. Furthermore, differential activation of SMA was significantly greater for MD-NSSI compared to MD-only. Relatively increased differential IFG activation in MD-only was observed when compared against HC (see Table 4). 

Exploratory analyses investigating relations between imaging findings and behavioral data from the Go/No Go task during fMRI in the patients revealed opposite correlations with error rates for MD-only versus MD-NSSI, particularly in the dACC. Data are presented in the Appendix A (see Appendix A).

## 4. Discussion

We investigated two clinical groups of adolescent MD patients with and without comorbid NSSI, and HC, using an established Erikson flanker Go/No Go paradigm during fMRI to examine the neural signaling of failed response inhibition. By implementing these two clinical samples in one study design, our investigation enabled us to disentangle specific neural patterns in brain regions relevant for failed response inhibition associated with NSSI. In contrast to the significantly enhanced reactivity in neural networks in all MD patients, we observed relatively blunted neural activation of these regions in the MD patients with comorbid NSSI. This might represent a correlate of impaired inhibitory control in the MD patients with NSSI, while neural reactivity to error processing in depressed adolescents without NSSI (MD-only) was increased not only compared to the other clinical group but also compared to healthy controls. Impaired inhibitory control (also often assessed by Go/No Go tasks) has also been reported from a systematic review and meta-analysis of adolescents and young adults with self-harm [51], which is in line with the findings presented in this study.

Behaviorally, the rate of commission errors for incongruent No Go trials did not significantly differ between MD-only and HC groups. Intact or even pronounced neural error processing, together with the equal or even smaller rate of commission errors in MD-only compared to HC, is in line with previous observations [24,52]. Hereby, the high accuracy is frequently accompanied by relatively slower reaction times and has been interpreted as a trade-off between accuracy and speed in depressed patients [53]. By contrast, the MD-NSSI group committed significantly more commission errors than HC and, as a trend, also more than MD-only. Furthermore, reaction times for commission errors were significantly increased compared to MD-only. Both phenomena are in line with impaired response inhibition, and potentially less cognitive control, in MD-NSSI patients. However, it has to be noted that a recent review pointed out that a robust pattern showing alterations in neural systems of cognitive control deficits in NSSI has not been established [54].

Neurofunctionally, we observed the expected differential activation in previously reported networks related to response inhibition and commission errors [27,55,56,57]. In particular, differential activation of the dACC, SMA, bilateral anterior insula, and bilateral inferior frontal gyrus was observed for each of the three groups (see Appendix A). Comparing the three groups in our whole-brain analysis, we observed significantly decreased differential neural activation related to errors of commission (No Go errors) in MD-NSSI for the left IFG and midline dACC compared to MD-only. Decreased differential neural activation of the right IFG and middle frontal gyrus was evident for MD-NSSI patients when compared to HC. This effect of blunted differential neural activation in the MD-NSSI group, together with the increased number of errors of commission in MD-NSSI, can be interpreted as a correlate of impaired inhibitory control, as, e.g., discussed in Case et al. [58] and Schreiner et al. [59]. By contrast, MD-only patients presented increased differential neural activation of the dACC/IFG network for errors of commission when compared against MD-NSSI and even to HC. 

In particular, the right-lateralized IFG has been linked to the specific executive function of response inhibition in neuroimaging studies with the Go/No Go paradigm, stop-signal tasks, or tasks involving cognitive set switching or interference suppression [19,29,60,61,62]. Graded neural activation in the IFG system was found to increase with increased demands for response control [63], allowing for the interpretation that lower activation might represent either a lower demand or perhaps also a lower ability for response control. Other neuroimaging studies showed co-activation of the right IFG during response inhibition, but also during other tasks, such as cognitive set switching [61] or interference suppression [62]. These findings indicate a broader role for this region in attentional and cognitive control [64]. Furthermore, differences in cognitive control may relate to differences in impulsivity [18], which may explain why our MD-NSSI patients showed an altered neural response for this region.

Regarding the dACC, previous studies consistently linked neural activations within this region with performance and error monitoring [56,65,66,67,68]. Moreover, according to [69], the dACC is involved in directing attention toward task-relevant stimuli and in maintaining associations between actions and their outcomes. Thus, activation of the dACC was suggested to serve to improve task performance through the modulation of control over the motor system and via allocating the capacities of different competing neural systems [70]. Several electrophysiological and fMRI studies [24,35,36,37,38] have already observed increased neural activation of this region during error processing in patients with major depression, as in the MD-only group investigated here. This may reflect a potential neural correlate of enhanced sensitivity to negative environmental cues or errors in major depression [71,72], which is more present in MD-only patients than in MD-NSSI patients (see Table 3). 

Recent neuroimaging studies with fMRI using cognitive and emotional tasks [48,73,74,75] and a meta-analysis [76] provide further evidence for a distinct neurobiology regarding brain functioning in patients with NSSI. Involving a similar network as in the current study on error processing, altered neural reactivity during reward and emotion processing was observed for the rostral ACC and the left IFG extending into the insula. Further studies could try to elucidate whether altered reactivity within this network could serve as a more global biomarker for clinical interventions and in developing diagnoses.

Regarding the limitations of our study, in particular, the relatively small sample compromises the strength and the generalizability of our data, and, therefore, the present results await empirical replication within larger samples. Moreover, the use of psychostimulant medication in one patient in the MD-only group may have potentially altered the neural activations relevant for error processing. However, control for this potential confounder confirmed that the overall pattern of group differences did not change substantially. Similarly, control calculations to exclude confounders due to motor activity still yielded similar results, with only smaller cluster sizes compared to the original analyses. Conceptually, it is important to note that the type of impulsivity that is captured by the present Go/No Go paradigm is situated in the cognitive domain, with impulsivity operationalized as the inability to inhibit behavioral impulses, as an important component of executive functions [77]. Furthermore, impulsivity was measured in a non-emotionally laden situation, and it would be interesting to investigate response inhibition in a more emotionally laden experimental situation as the driving force behind its failure.

By examining neural responses to errors in depressed adolescents with and without NSSI, we aimed to improve our neurobiological understanding of NSSI and hope to stimulate improvements in diagnostic procedures and treatment outcomes or monitoring. We observed an attenuated neural response in brain regions previously related to error processing and response inhibition, along with poorer task performance in depressed patients with comorbid NSSI. This pattern may represent a possible correlate for the clinically observed impaired inhibitory control in NSSI. Considering that there are very few neuroimaging data on similar tasks in NSSI, the replication and validation of our findings is needed in larger samples before these potential biomarkers can be translated into clinical practice.

## Figures and Tables

**Figure 1 biomedicines-10-03188-f001:**
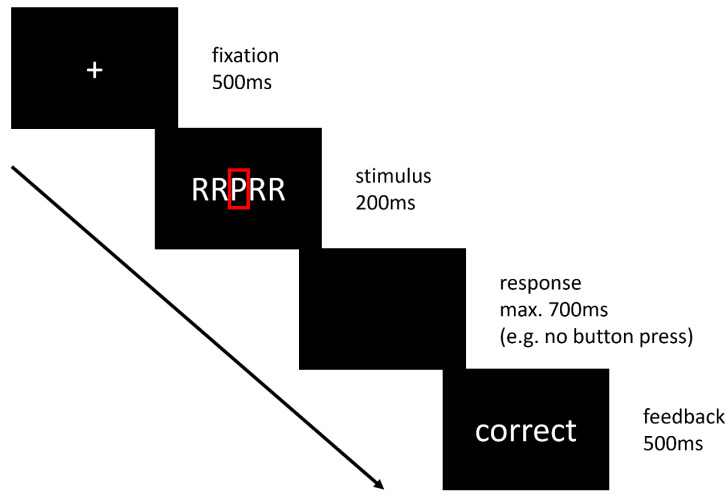
Erikson flanker Go/No Go paradigm as used during fMRI. The figure shows a sequence of an incongruent No Go trial as an example, starting from fixation until feedback. The No Go target letter ‘P’ is highlighted by a red rectangle only for demonstrational purposes (not shown during the real fMRI experiment). Durations of each trial segment are listed in milliseconds (ms).

**Figure 2 biomedicines-10-03188-f002:**
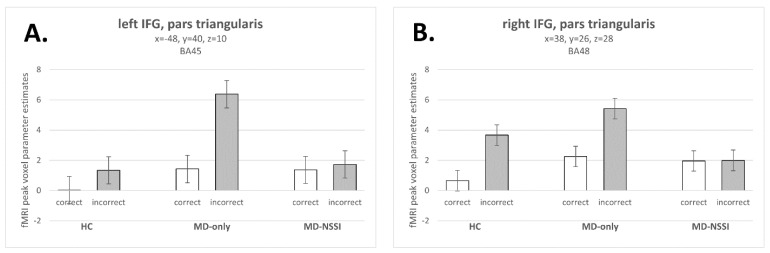
Neural activations of inferior frontal gyrus (IFG) for incorrect and correct incongruent No Go trials ((**A**). Left IFG; (**B**). Right IFG) in depressed adolescents with comorbid non-suicidal self-injury (MD-NSSI; n = 14), adolescents with major depression (MD-only; n = 12), and healthy controls (HC; n = 14). Bar charts depict fMRI peak voxel parameter estimates extracted from significant clusters located in left (MD-only minus MD-NSSI) and right (HC minus MD-NSSI) pars triangularis of IFG; error bars depict standard error of the mean. x, y, z: MNI coordinates provided in mm; BA = Brodmann area.

**Table 1 biomedicines-10-03188-t001:** Socio-demographic and questionnaire data in healthy controls (HC), adolescents with major depression (MD-only), and depressed adolescents with comorbid non-suicidal self-injury (MD-NSSI). Group differences were calculated with ANOVAs, followed by Bonferroni-corrected post-hoc comparisons at an alpha level of *p* ≤ 0.05.

				ANOVAs with Corresponding Post-Hoc Comparisons							
	HC	MD-Only	MD-NSSI			MD-NSSI vs. MD-Only		HC vs. MD-Only		HC vs. MD-NSSI	
	Mean (SD)	Mean (SD)	Mean (SD)	F	*p*	Average Difference	p_bonf_	Average Difference	p_bonf_	Average Difference	p_bonf_
**Demographics**											
N	14	13	14	-	-						
Female	n = 11	n = 10	n = 11	-	-						
Male	n = 3	n = 3	n = 3	-	-						
age [years]	14.4 (1.8)	16.2 (1.4)	15.4 (1.9)	3.349	0.046	−0.797	0.721	−1.725	0.041	−0.929	0.494
**Psychometry**											
IQ	109.6 (10.6)	101.8 (11.8)	101.3 (13.2)	2.093	0.137	-	-	-	-	-	-
CDRS-R score	21.4 (2.6)	56.4 (7.4)	51.4 (13.6)	59.29	<0.001	−4.956	0.498	−34.956	<0.001	−30.000	<0.001
BDI-II score *	2.9 (3.6)	27.2 (10.7)	21.7 (12.2)	23.988	<0.001	−5.452	0.462	−24.31	<0.001	−18.857	<0.001
Number of NSSI events last year	0	0	93.3 (225.9)	-	-						

Abbreviations: ANOVA: analysis of variance; SD: standard deviation; F: F-value; *p*: *p*-value; pbonf: Bonferroni-corrected *p*-value; n: sample size; []: corresponding unit; IQ: intelligence quotient; CDRS-R score: score on Children’s Depression Rating Scale—Revised; BDI-II score: score on Beck Depression Inventory—Second Version; *: BDI-II data from one subject in the MD-only group were missing (n = 12).

**Table 2 biomedicines-10-03188-t002:** Performance data in healthy controls (HC; n = 14), adolescents with major depression (MD-only; n = 13), and depressed adolescents with comorbid non-suicidal self-injury (MD-NSSI; n = 14). Group differences were calculated with ANOVAs, followed by Bonferroni-corrected post-hoc comparisons at an alpha level of *p* ≤ 0.05.

				ANOVAs with Corresponding Post-Hoc Comparisons							
	HC	MD-Only	MD-NSSI			MD-NSSI vs. MD-Only		HC vs. MD-Only		HC vs. MD-NSSI	
	Mean (SD)	Mean (SD)	Mean (SD)	F	*p*	Average Difference	p_bonf_	Average Difference	p_bonf_	Average Difference	p_bonf_
**Incongruent No Go**											
Number of commission errors	9.6 (6.5)	11.6 (6.9)	18.0 (11.6)	3.517	0.040	6.385	0.193	−1.973	1.000	−8.357	0.046
Reaction time [ms]	448.5 (57.9)	390.4 (53.8)	457.3 (67.8)	4.562	0.017	−66.877	0.024	58.138	0.058	−8.739	1.000
**Incongruent Go**											
Number of errors	2.8 (2.0)	3.5 (3.7)	4.3 (4.1)	0.686	0.510	-	-	-	-	-	-
Reaction time incorrect [ms]	470.4 (39.4)	393.5 (43.3)	412.8 (47.3)	9.883	<0.001	19.259	0.917	76.88	<0.001	57.621	0.010
Number of correct responses	21.1 (11.0)	23.3 (11.0)	20.4 (10.3)	0.263	0.771	-	-	-	-	-	-
Reaction time correct [ms]	468.0 (43.0)	410.4 (36.0)	427.0 (46.9)	6.685	0.003	16.614	0.946	57.643	0.003	41.029	0.044

Abbreviations: ANOVA: analysis of variance; SD: standard deviation; F: F-value; *p*: *p*-value; pbonf: Bonferroni-corrected *p*-value; n: sample size; []: corresponding unit; ms: milliseconds.

**Table 3 biomedicines-10-03188-t003:** Significant (*p* ≤ 0.05; minimum cluster size of 183 contiguously significant voxels at *p* < 0.005, uncorrected) between-group differences in differential neural activations from the contrast ‘incorrect minus correct incongruent No Go trials’ corresponding to neural error signaling. Depressed adolescents with comorbid non-suicidal self-injury: MD-NSSI (n = 14); adolescents with major depression: MD-only (n = 12; no first-level contrast in 1 patient with zero No Go errors); healthy controls: HC (n = 14).

	BA	Anatomic Label	L/R/M	Cluster Size	Z	MNI		
				*k (Vx)*		*x*	*y*	*z*
**MD-NSSI < MD-only**	32	dACC	M	183	3.80	10	22	38
	45	IFG, pars triangularis	L	427	3.62	−48	40	10
	48	IFG, pars opercularis	L	348	4.34	−38	4	28
	40	supramarginal gyrus	L	498	3.13	−64	−34	30
	2	inferior parietal cortex	L	#	3.53	−42	−32	40
**HC < MD-only**	48	anterior insula	L	1238	4.45	−38	−2	18
	48	postcentral gyrus	L	#	3.88	−48	−6	18
	3	postcentral gyrus	L	#	3.82	−56	−16	40
	40	inferior parietal cortex	L	#	3.23	−40	−34	38
		precuneus	L	438	4.02	−14	−50	50
	37	cerebellum	L	189	6.63	−20	−42	−26
**MD-NSSI < HC**	45	middle frontal gyrus	R	449	3.62	44	34	24
	48	IFG, pars triangularis	R	#	3.27	38	26	28
	48	IFG, pars opercularis	R	#	3.24	44	16	30
	48	supramarginal gyrus	R	366	3.63	58	−42	26
	40	inferior parietal cortex	L	191	4.00	−42	−44	52
**HC < MD-NSSI**		precuneus	L	944	4.05	−12	−46	52

n: sample size; BA: Brodman area; L/R/M: left/right/midline; Z: z-score of standard normal distribution; k: number of voxels (Vx); MNI: Montreal Neurological Institute (x-, y-, z-coordinates are provided in mm); mm: millimeter; dACC: dorsal anterior cingulate cortex; IFG: inferior frontal gyrus; #: maximum part of cluster above.

**Table 4 biomedicines-10-03188-t004:** Significant (*p* ≤ 0.05; minimum cluster size of 183 contiguously significant voxels at *p* < 0.005, uncorrected) between-group differences in differential neural activations from the contrast ‘incorrect minus correct incongruent Go trials’ corresponding to neural error signaling. Depressed adolescents with comorbid non-suicidal self-injury (MD-NSSI; n = 14); adolescents with major depression (MD-only; n = 13); healthy controls (HC; n = 14).

	BA	Anatomic Label	L/R/M	Cluster Size	Z	MNI		
				*k (Vx)*		*x*	*Y*	*z*
**MD-NSSI < MD-only**	45	IFG, pars triangularis	L	232	4.12	−48	26	10
	45	IFG, pars triangularis	R	199	3.11	54	32	4
	48	IFG, pars opercularis	R	#	3.53	50	14	6
**MD-only < MD-NSSI**	6	SMA	M	297	4.02	2	−4	56

**HC < MD-only**	45	IFG, pars triangularis	L	499	4.60	−48	30	12
	48	anterior insula	L	#	3.15	−40	14	6

n: sample size; BA: Brodman area; L/R/M: left/right/midline; Z: z-score of standard normal distribution; k: number of voxels (Vx); MNI: Montreal Neurological Institute (x-, y-, z-coordinates are provided in mm); IFG: inferior frontal gyrus; #: maximum part of cluster above; SMA: supplementary motor area.

## Data Availability

Not applicable.

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
