# Peer review of "Neural Signatures of Error Processing in Depressed Adolescents with Comorbid Non-Suicidal Self-Injury (NSSI)"

_biomedicines, 2022, doi:10.3390/biomedicines10123188_

Round 1

Reviewer 1 Report

The authors present a well designed paper in a difficult to study population. The paper is well written with a sound structure. I have the following few comments

1. Introduction line 39-40. I am not convinced that this reflects clinical reality, that self-harm and borderline personality syndrom rarely occur together. The authors have chosen to look at NSSI in relation to depression, but the relationship between NSSI and personality syndrom need not be downplayed, and authors should be careful to not selectively cite the literature to fit the structure of the paper

2. Impulsivity is not a unitary construction, the type of impulsivity that is captured with Go /NoGo paradigm should be described

3. Line 65-69: SMA has also been shown to be an equally  important area in response inhibition, should may be mentioned here. 

Author Response

Reviewer 1:

The authors present a well designed paper in a difficult to study population. The paper is well written with a sound structure. I have the following few comments

Response: Thank you very much for the kind and considerate highlighting of the strengths of our work and also the highly valuable remarks that have helped to further improve the manuscript.

  1. Introduction line 39-40. I am not convinced that this reflects clinical reality, that self-harm and borderline personality syndrom rarely occur together. The authors have chosen to look at NSSI in relation to depression, but the relationship between NSSI and personality syndrom need not be downplayed, and authors should be careful to not selectively cite the literature to fit the structure of the paper

Response 1: Thank you for pointing this out. We toned down our statement to: „However, in earlier adolescence, it has been shown that particularly mood disorders and NSSI often co-exist [9,10]. While NSSI is seen as an important predictor for the development of borderline personality disorder in early adulthood, only a minority of adolescent patients with self-injurious behavior fulfil the criteria for the diagnosis around the onset of NSSI behavior [11]“.

  1. Impulsivity is not a unitary construction, the type of impulsivity that is captured with Go /NoGo paradigm should be described

Reponse 2: We agree with you that impulsivity is not a unitary construct, and its definition depends on the perspective, be it more psychopathologically motivated, or more from a cognitive standpoint, at the very least. This has recently been nicely reviewed by Bakhshani (2014),  who comes to the conclusion that “(…) the problem of giving a single definition for impulsivity still exists.” If we had to categorize the type of impulsivity that is captured with the present Go/NoGo task, we would stress a cognitive viewpoint where impulsivity is the inability to inhibit behavioral impulses as an important component of executive functions. We made an addendum to the limitations sectionto make the reader aware of this issue and direct readers’ attention to the recent article by Bakhshani (2014) mentioned above. Motivated by your comment we now also mention, that the translation of the present paradigm to more emotionally laden situations may appear as an interesting avenue for future research.

  1. Line 65-69: SMA has also been shown to be an equally important area in response inhibition, should may be mentioned here. 

Reponse 3: Thank you for making us aware of this issue. Functional relevance of the SMA in response inhibition has been made explicit at the corresponding place.

Reviewer 2 Report

This is an interesting study showing the neural features of error processing in NSSI-depressed adolescents. Indeed, the self-harm with impulsivity and depression in young people are less un-noticed, and the current data are a nice contribution to the area. However, the authors might pay attention to the following points.

1. Although the study was focused on adolescents, their affective or personality disorder tendencies should be clearer in this design. For instance, bipolar or borderline/antisocial personality pathologies should be controlled thereafter.

2. The disease history of patients should be described more clearly since it might be related to the neuroimaging features, and help to illustrate the disease path process. The frequency of parasuicidal behaviors might be exciting in this design.

3. The behavioral parameters recorded during the neuroimaging test, for instance, the number of errors, RT or others, might also be related to the neuroimaging results.

4. There was an age-group difference (Table 1, age, p = 0.046) between healthy controls, MD, and MD-NSSI, therefore, the authors might recruit some more healthy controls to balance the age means, since that will be the easiest way to overcome the shortage.

5. How about the personality trait measures in these young participants, especially sensation seeking, which relates to impulsivity?

Author Response

Reviewer 2

This is an interesting study showing the neural features of error processing in NSSI-depressed adolescents. Indeed, the self-harm with impulsivity and depression in young people are less un-noticed, and the current data are a nice contribution to the area. However, the authors might pay attention to the following points.

Response: Thank you very much for the encouraging feedback and for the helpful remarks to improve the manuscript.

  1. Although the study was focused on adolescents, their affective or personality disorder tendencies should be clearer in this design. For instance, bipolar or borderline/antisocial personality pathologies should be controlled thereafter.

Response 1: Thank you for bringing this to our attention. Participants’ diagnoses were assessed using the German version of the clinical interview Schedule for Affective Disorders and Schizophrenia for School-Age-Children-Present and Lifetime (Kiddie-SADS-PL) for DSM-IV diagnoses (Delmo et al.2000). Hereby, bipolar disorder as well as schizophrenia would have been detected and would have been treated as an exclusion criterion. This is now mentioned in the text (line 112). Regarding borderline or antisocial personality pathologies, the Kiddie-SADS permitted diagnosis of BPS in only one subject of the MD-NSSI group (line 100). Dimensional measures informing about symptom severity of borderline or antisocial personality pathologies were not applied.

  1. The disease history of patients should be described more clearly, since it might be related to the neuroimaging features, and help to illustrate the disease path process. The frequency of parasuicidal behaviors might be exciting in this design.

Response 2: Following your suggestion to present more information on parasuicidal behaviors, we now present the following data assessed with the semi-structured Self-injurious thoughts and behaviors interview German (SITBI-G; Fischer et al. 2014) in section 3.1 Demographic data and psychometric measurements: “According to the SITBI-G, 1 patient in the MD-only group and 4 patients in the MD-NSSI group reported suicide attempts with a total number of 1 (MD-only) and a mean number of 3.3 in MD-NSSI (SD=3.9), respectively. Suicide plans were reported by the same patient in the MD-only group at 1 occasion within the past year. In the MD-NSSI group, 6 patients reported suicide plans at a mean rate of 3.3 (SD=2.8) episodes within the last year. None of the participants in the HC group reported suicide attempts or plans. Suicidal thoughts were reported by participants from each group (HC: n=2; MD-only: n=12; MD-NSSI: n=13) at mean rates from 1 episode within the past year (HC group), over 2.8 (SD=3.0; MD-only) to 13.8 (SD=31.1; MD-NSSI) episodes in the past year.” No meaningful correlations with neuroimaging data were found.

  1. The behavioral parameters recorded during the neuroimaging test, for instance, the number of errors, RT or others, might also be related to the neuroimaging results.

Response 3:

Following the suggestion, we examined significant associations between neural activations and behavioral responses (number of commission errors and reaction times) during fMRI in the patients The results are presented in the supplementary material.

Differential fMRI parameter estimates of significant peak voxel activations from between-group comparisons were extracted for correlation analyses, namely from dACC and IFG as key structures of neural error processing. We observed a significant negative correlation between the individual number of commission errors and individual neural activations of the contrast `incorrect minus correct NoGo trials` within the dACC in MD-only (r=-0.653; p=*0.021). After correcting for one outlier in MD-NSSI with an extraordinary high number of commission errors (n=48), a reversed, significant correlation was seen in MD-NSSI (r=0.574; p=*0.040). No significant correlations were detected regarding reaction times. Corresponding graphs are shown in figure S1.

A relationship between differential fMRI parameter estimates and commission errors with corresponding reaction times in NoGo trials was also apparent in the left inferior frontal gyrus (IFG, pars opercularis). However, differential activities were significantly correlated only in MD-only with number of commission errors/reaction time (r=-0.700; p=*0.011/r=0.645; p=*0.023), while opposite correlations were not significant in MD-NSSI.

  1. There was an age-group difference (Table 1, age, p = 0.046) between healthy controls, MD, and MD-NSSI, therefore, the authors might recruit some more healthy controls to balance the age means, since that will be the easiest way to overcome the shortage.

Response 4: Unfortunately, due to a complete change of the MR scanner, increasing the group of healthy controls in order to better balance age means is no longer possible with the same scanner. The small difference of age means of 1.6 years between MD-only and HC groups was statistically significant only due to the rather small age range of each sample. As the main findings of the paper relate to the differences between the two patient groups, where the  age difference of around 0.6 years was even smaller and not significant, we do not think that age has an impact.

  1. How about the personality trait measures in these young participants, especially sensation seeking, which relates to impulsivity?

Response 5: As the data presented here were collected as part of a larger experimental setup, where participants underwent a Cyberball task and unpleasant electrical stimulation (see lines 137-1394), assessments of trait measures focused particularly on that Cyberball task. As a result, measures of sensation seeking or related traits were not taken.

Reviewer 3 Report

Thank you for the opportunity to review this quality work. Authors presented a novel investigation aimed at examining neural responses to errors in depressed adolescents with and without non-suicidal self injury (NSSI). Robust methods were used to test the study hypothesis, and the results are well presented and discussed. The study has the potential to fill a significant gap in the literature and can indeed improve our neurobiological understanding of NSSI.

Author Response

Thank you for the opportunity to review this quality work. Authors presented a novel investigation aimed at examining neural responses to errors in depressed adolescents with and without non-suicidal self injury (NSSI). Robust methods were used to test the study hypothesis, and the results are well presented and discussed. The study has the potential to fill a significant gap in the literature and can indeed improve our neurobiological understanding of NSSI.

 Response: Thank you very much for the kind and considerate highlighting of the strengths of our work and the encouraging feedback!

Round 2

Reviewer 2 Report

No further comments